# Seed Germination Ecology of *Chenopodium album* and *Chenopodium murale*

**DOI:** 10.3390/biology11111599

**Published:** 2022-11-01

**Authors:** Ram Swaroop Bana, Vipin Kumar, Seema Sangwan, Teekam Singh, Annu Kumari, Sachin Dhanda, Rakesh Dawar, Samarth Godara, Vijay Singh

**Affiliations:** 1ICAR-Indian Agricultural Research Institute, New Delhi 110 012, India; 2Eastern Shore Agricultural Research and Extension Center, Virginia Tech, Painter, VA 23420, USA; 3Department of Crop, Soil & Environmental Sciences, Auburn University, Auburn, AL 36849, USA; 4Department of Agronomy, Kansas State University, Manhattan, KS 66506, USA; 5ICAR-Indian Agricultural Statistics Research Institute, New Delhi 110 012, India

**Keywords:** osmotic stress, photoperiod, pH, salinity stress, soil reaction, temperature, weed biology

## Abstract

**Simple Summary:**

*Chenopodium album* L. and *Chenopodium murale* L. species of weeds are significant menaces in field crops and are reported to have evolved resistance against numerous herbicides across the world. Studying their germination biology can help in devising non-chemical management strategies. We investigated the germination behaviour of these two weed species under variable temperatures, light conditions, salinity, water stress, and pH. The results from this study reveal that these two weed species can germinate over a wide range of environmental conditions, which might help them to spread and establish in new ecologies.

**Abstract:**

*Chenopodium album* L. and *Chenopodium murale* L. are two principal weed species, causing substantial damage to numerous winter crops across the globe. For sustainable and resource-efficient management strategies, it is important to understand weeds’ germination behaviour under diverse conditions. For the germination investigations, seeds of both species were incubated for 15 days under different temperatures (10–30 °C), salinity (0–260 mM NaCl), osmotic stress (0–1 MPa), pH (4–10), and heating magnitudes (50–200 °C). The results indicate that the germination rates of *C. album* and *C. murale* were 54–95% and 63–97%, respectively, under a temperature range of 10 to 30 °C. The salinity levels for a 50% reduction in the maximum germination (GR_50_) for *C. album* and *C. murale* were 139.9 and 146.3 mM NaCl, respectively. Regarding osmotic stress levels, the GR_50_ values for *C. album* and *C. murale* were 0.44 and 0.43 MPa, respectively. The two species showed >95% germination with exposure to an initial temperature of 75 °C for 5 min; however, seeds exposed to 100 °C and higher temperatures did not show any germination. Furthermore, a drastic reduction in germination was observed when the pH was less than 6.0 and greater than 8.0. The study generated information on the germination biology of two major weed species under diverse ecological scenarios, which may be useful in developing efficient weed management tactics for similar species in future agri-food systems.

## 1. Introduction

Globally, around 250 plant species belong to the Chenopodium genus of the Chenopodiaceae family, of which 25 are acknowledged as weeds [1]. Amongst these, *Chenopodium album* L. (common lambsquarters) and *Chenopodium murale* L. (nettleleaf goosefoot) are the two foremost weed species. Both species are annual, broad-leaved herbaceous weedy plants. *C. album* is one of the top five extensively growing plant species all over the world [2], largely owing to its copious seed-producing proficiency, yielding up to 70,000 seeds per plant [3]. Likewise, *C. murale* can produce >24,000 seeds per plant [4]. The high seed longevity of the two species significantly contributes to their severity [5].

Both *C. album* and *C. murale* are temperate and sub-temperate weeds; however, they can infest the winter crops of tropics and subtropics, as well. The two species have been reported to infest over 25 crops in different ecologies of the world, including wheat (*Triticum aestivum* L.), maize (*Zea mays* L.), potato (*Solanum tuberosum* L.), cauliflower (*Brassica oleracea* L.), garlic (*Allium sativum* L.), onion (*Allium cepa* L.), and spinach (*Spinacia oleracea* L.), and cause substantial yield damage based on the scale of infestation [6,7,8]. Seeds of *C. album* and *C. murale* reaped with crop plants deteriorate the product quality and reduce the economic value of the crop produce [9]. Along with various indirect losses, both species can result in direct crop losses through the release of several allelochemicals, and their allelopathic effects have been reported on sunflower, tomato, mustard, and a few other crop species [7,10,11]. Furthermore, *C. album* also serves as an alternate host for plentiful plant pathogens (*Stagonospora atriplicis, Polymyxa betae,* Yellow Vein Virus, Beet Yellows Virus, Peanut Stunt Cucumovirus, Prunus Necrotic Ring Spot Virus), insects (*Pemphigus betae*), and nematodes (*Ditylenchus destructor, Meloidogyne incognita*) [12,13,14,15,16,17,18,19]. Besides crop fields, both species propagate luxuriantly on wastelands, pastures, and uncultivated lands. Pollens of *C. album* are known to cause hay fever [20]. Furthermore, owing to high nitrate and oxalic acid concentrations in *C. album,* its excessive consumption results in severe health adversities and fatalities in animals [21,22,23].

Despite being a major weed of winter crops, both *C. album* and *C. murale* were previously not major menaces, as it was relatively easy to manage these species due to the availability of effective herbicides for their control. However, in the recent past, the emergence of herbicide resistance in *C. album* in many countries has raised concerns over the management of this weed [24]. Globally, *C. murale* has been reported resistant to herbicides of triazines and triazinone, substituted ureas, nitrile, pyrazole, imidazolinone, and the auxin mimic group [25,26,27,28,29,30,31,32,33,34]. Therefore, for effective management of these weed species under the changing scenario, it is important to understand diverse aspects of their germination behaviour to develop non-herbicidal control options. Early knowledge about the effect of different factors on the germination pattern of a weed species can assist in devising pragmatic control strategies [35,36]. Hence, studying the effects of various ecological factors on the germination of *C. album* and *C. murale* seeds would be helpful in developing management approaches by discouraging or stimulating their emergence at times when seedlings can be effortlessly controlled. Furthermore, seed germination is a complex phenomenon and is regulated by various elements such as seed dormancy and edaphic, climatic, and environmental factors [36]. So far, no systematic studies have been conducted to explore the effect of different factors on the germination behaviour of *C. album* and *C. murale*. There is a lack of basic information on the biology of these two species of global importance. Therefore, keeping the knowledge gap in view, the present study was undertaken to investigate the effects of diverse ecological factors such as temperature, photoperiod, salinity levels, water stress, and pH on the germination of *C. album* and *C. murale.*

## 2. Materials and Methods

### 2.1. Seed Description

Seeds were harvested from wheat fields in April 2019 from 15 to 20 mature *C. album* and *C. murale* plants growing at the Research Farm of Division of Agronomy, ICAR-Indian Agricultural Research Institute (IARI), New Delhi, situated at latitude 28°4′ N, longitude 77°12′ E, and 228.6 m MSL altitude. The annual rainfall of Delhi is 652 mm (70–80% of which is received during July–September and the rest during October–May) and the annual pan evaporation (PE) is ~850 mm. The soil of the site belongs to Inceptisols, having a sandy loam texture with a slightly alkaline pH (7.8 in 1:2.5 soil:water). The soil organic carbon of the seed collection site was 0.48%, and KMnO_4_ oxidizable N, 0.5 N NaHCO_3_ extractable P, and 1 N NH_4_OAc-extractable K were 181.3, 16.7, and 244 kg ha^−1^, respectively.

Seeds collected from all the plants of individual species were mixed to make a composite sample for each species. After harvesting, seeds were dried under shade for 3–4 days, and afterwards, the seeds were rubbed by hand to remove their thin layer of husk. The seeds were then stored in polythene bags at 25 °C until the start of the germination studies.

### 2.2. General Seed Germination Procedure

Seed germination studies were carried out in the Division of Seed Science and Technology, ICAR-IARI, New Delhi. For the germination assessment, 25 seeds were placed in plastic Petri plates (diameter 10 cm) containing two Whatman no. 1 filter papers (Figure 1). Petri plates were whipped with 75% ethanol before placing the seeds and filter papers to prevent any microbial contamination in the Petri plates during the incubation period. Filter papers were moistened with 5 mL double-distilled water or test solutions. These Petri plates were incubated at 20 °C unless otherwise specified in 12/12 h light/dark environments (except for the experiments focused on light duration effects on germination of seeds, where some Petri plates were placed in completely dark settings throughout the study). Seeds were considered germinated when the radicles emerged ≥2 mm in length. Seeds were incubated for 15 days and the germination count was observed every alternate day (days 3, 5, 7, 9, 11, 13, and 15). Germinated seeds were removed from the Petri plates after finishing the germination count.

### 2.3. Effect of Temperature and Light on Germination

To find the optimum temperature conditions for the germination of *C. album* and *C. murale*, seeds were placed at different temperatures, i.e., 10, 15, 20, 25, and 30 °C in 12/12 h light/dark scenarios. These temperature ranges were selected to match the average temperature of different countries across the globe.

To study the photoperiod effect, Petri plates were incubated in 12/12 h light/dark situations and in completely dark settings (the Petri plates were covered with three layers of 0.016 mm thick aluminium foils to restrict the exposure of the seeds to the light) at 20 °C. The non-germinated seeds after 15 days of incubation in completely dark conditions were then exposed to 12/12 h light/dark environments to understand whether the absence of light was accountable for the lack of germination.

### 2.4. Effect of Salt Stress on Germination

To understand the salt stress effects on the germination of *C. album* and *C. murale,* salt solutions of different concentrations (0, 0.03, 0.06, 0.1, 0.14, 0.18, 0.22, and 0.26 M NaCl) were applied for moistening the filter papers instead of distilled water. These solutions were prepared by dissolving 0, 1.755, 3.51, 5.85, 8.19, 10.53, 12.87, and 15.21 g of NaCl in double-distilled water and making the final volume 1 litre. To detect whether the germination inhibition at higher salt concentrations was due to physiological drought caused by the high salt concentration or due to the toxic effect of saline ions, the seeds that did not germinate at higher salt concentrations were rinsed for 5 min with running water and re-incubated for the germination studies without salt stress.

### 2.5. Effect of Osmotic Stress on Germination

To evaluate the influence of osmotic stress on the germination of *C. album* and *C. murale* seeds, incubation of the seeds was carried out in the solutions of various water potentials, i.e., 0, −0.1, −0.2, −0.4, −0.6, −0.8, and −1 MPa. These solutions were prepared by dissolving 0, 45.8, 64.7, 91.5, 112.1, 129.5, and 144.9 g polyethylene glycol 8000 in 0.5 L distilled water [37]. After 15 days of incubation, the seeds which did not germinate in high osmotic potential solutions were washed thoroughly in running tap water and then incubated again for 15 days in normal distilled water-soaked filter paper.

### 2.6. Effect of pH on Germination

For detecting the germination sensitivity of *C. album* and *C. murale* seeds to pH, seeds were incubated at different pH levels (4, 5, 6, 7, 8, 9, and 10), and distilled water was used as a control (6.7 pH). These buffer pH solutions were prepared as described by [38].

### 2.7. Effect of Initial High Temperature on Weed Germination

For determining the effect of crop residue burning on the germination of *C. album* and *C. murale*, the seeds were heated for 5 min in an oven at different temperatures, i.e., 50°, 75°, 100°, 125°, 150°, 175°, and 200 °C. This temperature range was selected to simulate the temperature of the upper soil horizon during crop residue/vegetation burning. Seeds stored at room temperature were taken as a control. After heating, the seeds were incubated at 20 °C in 12/12 h light/dark conditions for 15 days.

### 2.8. Statistical Analysis

The experiments were set up in a randomised complete block design with three replications. All the experiments were repeated (second run) with a gap of one month after the completion of the first run of experiments. Data from both runs were analysed to check homogeneity. No differences were noticed in either run; therefore, the data from both runs were pooled. Germination data from different parameters were fitted to the best-fitting regression model using Sigma Plot 14.5 (Systat Software, San Jose, CA, USA). The three-parameter sigmoidal model was found to be the best fit for germination percentage data at varying levels of osmotic stress, salt concentration, and initial high temperature. The best-fitting three-parameter sigmoid equation was:f = a/[1 + exp(−(x − x_50_)/b]
where f is the germination at any particular (x) germination parameter value, a is the maximum germination percentage observed for a specific parameter, x_50_ is the particular parameter value required to inhibit maximum germination by 50%, and b is the slope for the curve. Data from temperature and pH parameters followed a normal distribution and were subjected to ANOVA, and the means were separated using Tukey’s HSD test at a 5% level of significance using JMP^®^ Pro 16 (SAS Institute Inc., Cary, NC, USA).

## 3. Results and Discussion

### 3.1. Effect of Temperature and Light on Germination

The effect of temperature on the germination of *C. album* seeds was found significant (*p* < 0.001). However, germination of *C. album* was greater than 50% for the entire temperature regime (10–30 °C). Maximum and minimum germination were observed at 20 °C (95%) and 10 °C (54%), respectively (Figure 2). Germination of *C. album* ranged between 75% and 78% at 15, 25, and 30 °C, but it was statistically similar among temperatures (Figure 2). Similarly, the effect of temperature was significant for the germination of *C. murale* seeds (*p* < 0.001). Germination of *C. murale* was 63–97% for the entire temperature regime (10–30 °C). Maximum and minimum germination were observed at 20 °C (97%) and 10 °C (63%), respectively (Figure 1). At 25 and 30 °C, germination of *C. murale* was 80% and 77%; however, both temperatures yielded similar results. At low temperatures (10 and 15 °C), the germination of *C. murale* was higher than that of *C. album*. The results suggest higher infestation of *C. murale* under late-season conditions in winter crops as compared to *C. album.* Furthermore, in timely planted scenarios, the focus on *C. album* control in the early season will be an effective approach, owing to reduced germination at lower temperatures and a likelihood of lesser infestation at later stages of crop ontogeny [39]. Therefore, the present findings will be helpful in devising ecological approaches to management of the two species, either by tweaking the planting period or by shifting the focus of weed management towards expected dominant species under the different scenarios.

Furthermore, in the photoperiod studies, seeds of both species did not germinate under completely dark scenarios, whereas under 12/12 h light/dark conditions, germination was >95%. When the non-germinated seeds of both species from completely dark conditions were incubated under 12/12 h light/dark environments, more than 90% germination was observed. The photoperiod for germination of *C. album* has also been previously reported [40,41]. This indicates that either burying the seeds deep with tillage or switching over to conservation agricultural systems (CAS) is required to manage the weed infestation, as a thick crop residue cover will have a shading effect and thereby restrict the penetration of light to the seeds and thus reduce weed infestation in modern CAS [42].

### 3.2. Effect of Salt Stress on Germination

The germination percentage at different salt concentrations for both species was fitted to a three-parameter sigmoidal curve (Figure 3). The fitted equation for *C. album* was {f = 94.08/[1 + exp(−(x − 139.89)/−22.98], r^2^ = 0.99}, and that for *C. murale* was {f = 90.96/[1 + exp(−(x − 146.31)/−20.32], r^2^ = 0.98}. Both species were able to germinate with a NaCl concentration from 0 to 220 mM. However, the germination percentage was less than 15% at the concentration of 180 mM NaCl and higher. Germination for *C. album* and *C. murale* was more than 80% and 75%, respectively, at a salt concentration of 100 mM NaCl. The GR_50_ (50% reduction in maximum germination) for *C. album* and *C. murale* was 139.9 and 146.3 mM NaCl, respectively (Table 1), which indicated a slightly higher tolerance of *C. murale* to saline soils as compared to *C. album*. Neither species germinated at the salt concentration of 260 mM NaCl. The non-germinated seeds at the 260 mM NaCl salt concentration were rinsed thrice with tap water and placed in distilled water. The germination percentage for both species was recorded as more than 90% after triple rinsing, which indicates that the saline conditions do not affect the seed viability of both species.

The salt tolerance ability of *C. album* and other species from the *Chenopodium* genus has been reported in previous studies [43,44,45,46]. Germination over a wide range of salt concentrations makes these two weed species able to adapt to saline soils.

### 3.3. Effect of Osmotic Stress on Germination

A three-parametric sigmoidal curve was fitted for the germination percentage of both weed species at different osmotic stress levels (Figure 4). The best-fitting equation for *C. album* was {f = 96/[1 + exp(−(x − 0.44)/−0.053], r^2^ = 0.99}, and that for *C. murale* was {f = 95.72/[1 + exp(−(x − 0.43)/−0.058], r^2^ = 0.99}. Osmotic stress levels for a 50% reduction in the germination of *C. album* and *C. murale* were 0.44 and 0.43 MPa, respectively. Both species germinated in osmotic stress of 0–0.6 MPa. Germination of *C. album* and *C. murale* was 70% and 62%, respectively, at osmotic stress of 0.4 MPa. Eslami (2011) also reported more than 65% germination of *C. album* seeds at osmotic pressure of 0.4 MPa. However, germination of both species reduced drastically at osmotic stress of 0.6 MPa, with germination of 3% and 2.7% for *C. album* and *C. murale*, respectively. Seeds from both species did not germinate at osmotic stress of 0.8 MPa and higher. The non-germinated triple-washed seeds placed in distilled water resulted in 94% and 95% germination for *C. album* and *C. murale*, respectively. This indicates that exposure of *C. album* and *C. murale* seeds to high osmotic stress conditions does not affect their viability. The germination of *C. album* and *C. murale* over a wide range of osmotic stress levels supports the presence of both these species in areas of varying soil moisture.

### 3.4. Effect of Initial Temperature on Germination

The germination percentage after exposure of seeds to an initial high temperature for 5 min for both species was fitted to a three-parameter sigmoidal curve (Figure 5). The fitted equation for *C. album* was {f = 98.33/[1 + exp(−(x − 80.15)/−1.67], r^2^ = 0.99}, and that for *C. murale* was {f = 96.67/[1 + exp(−(x − 79.30)/−0.86], r^2^ = 0.99}. The two studied species resulted in >95% germination up to exposure of 75 °C initial temperature. However, germination for both species decreased drastically at the exposure of more than 75 °C. Seeds exposed to 100 °C and higher temperatures did not show any germination for both species. From the fitted equation, GR_50_ for *C. album* and *C. murale* was found to be 80.2 and 79.3 °C, respectively.

The practice of burning crops and plant residues for clearing farmland is followed by farmers in several parts of the world. The burning of vegetation can increase the soil surface temperature up to 550 °C for a 6 min period [47]. However, the temperature decreases as the soil depth increases at a gradient of 100 °C per cm of soil depth [48]. Assuming these numbers, burning crop and plant residues can destroy the germination and viability of *C. album* and *C. murale* seeds in the top 5 cm of soil. However, we do not support the burning of crop residues due to its harmful effect on soil health and the environment.

### 3.5. Effect of pH on Germination

The effect of pH was significant for the germination of both species (*p* < 0.001). For *C. album*, maximum germination (97%) was noticed at control (distilled water with pH 6.7), followed by pH 7.0 (96%) (Figure 6). Germination of *C. album* dropped significantly with pH 6.0 (65%) and 8.0 (73%). There was a drastic reduction in the germination of *C. album* seeds when the pH decreased below 6.0 and increased above 8.0 (Figure 6).

In a similar manner, the maximum germination for *C. murale* was also witnessed at control (99%) and pH 7.0 (98%). The germination of *C. murale* reduced to 69% and 73% by decreasing the pH to 6.0 and increasing the pH to 8.0, respectively. When the pH decreased to 5.0 or lower, the germination of both species remained between 10% and 15%. Increasing the pH to 9.0 reduced the germination of both species below 10%. Furthermore, germination for both species was 0% at pH 10.0 across all replications and both runs.

## 4. Conclusions

The seeds of *C. album* and *C. murale* have the potential to germinate profusely in medium temperature ranges (15–30 °C); however, a significant reduction in germination at lower temperatures was observed in *C. album*. This suggests that under scenarios of late-season planting of winter crops, a lesser *C. album* infestation is expected. Furthermore, in timely planted scenarios, the focus on *C. album* control in the early season will be an effective approach—owing to reduced germination at lower temperatures, there is the likelihood of lesser infestation at later stages of crop ontogeny. Secondly, the lack of germination of both species in dark conditions suggests that either burying the seeds deep or switching over to conservation agricultural systems (CAS) is required to manage the weed infestation. A thick crop residue cover will have a shading effect and thereby restrict the penetration of light to the seeds, thus reducing weed infestation in modern CAS. Thirdly, germination of both species over wide ranges of osmotic pressure, salinity, and pH makes them able to adapt to problematic soils and under stress ecologies. However, a drastic reduction in the germination of both weed species at extremely low and high pH was observed. Such wide-ranging germination responses strongly suggest that further investigations are needed on more variable soil and climatic conditions, soil depth, crop establishment, and tillage systems in realistic on-farm environments. Moreover, future research should also focus on the effect of climatic, soil, and management variables on the response of different biotypes of these two weed species from diverse agro-climatic ecosystems.

## Figures and Tables

**Figure 1 biology-11-01599-f001:**
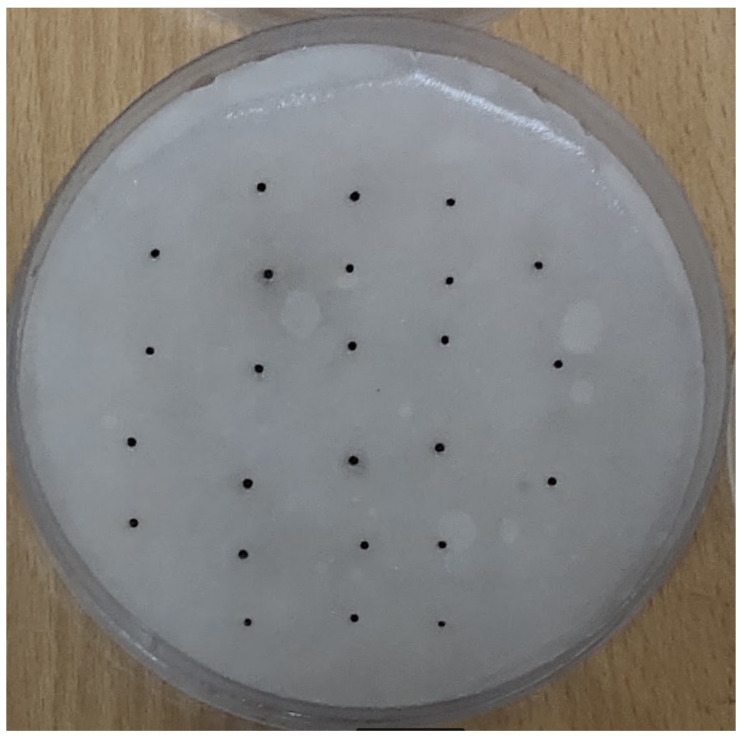
Arrangement of seeds for germination studies.

**Figure 2 biology-11-01599-f002:**
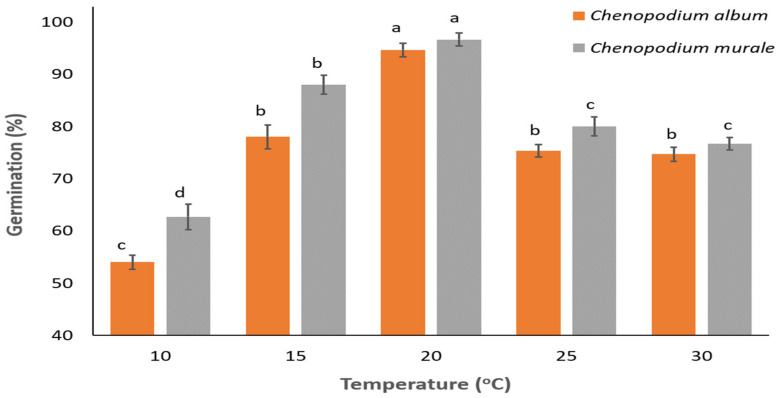
Effect of temperature on the germination of *Chenopodium album* and *C. murale.* Bars with different alphabets are statistically different for a particular species. The species were not compared to each other for germination at different temperatures.

**Figure 3 biology-11-01599-f003:**
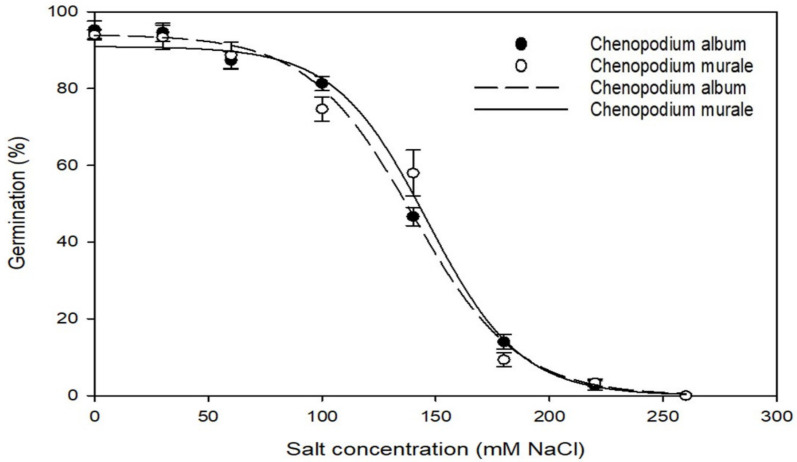
Effect of salt concentration on germination percentage of *C. album* and *C. murale*.

**Figure 4 biology-11-01599-f004:**
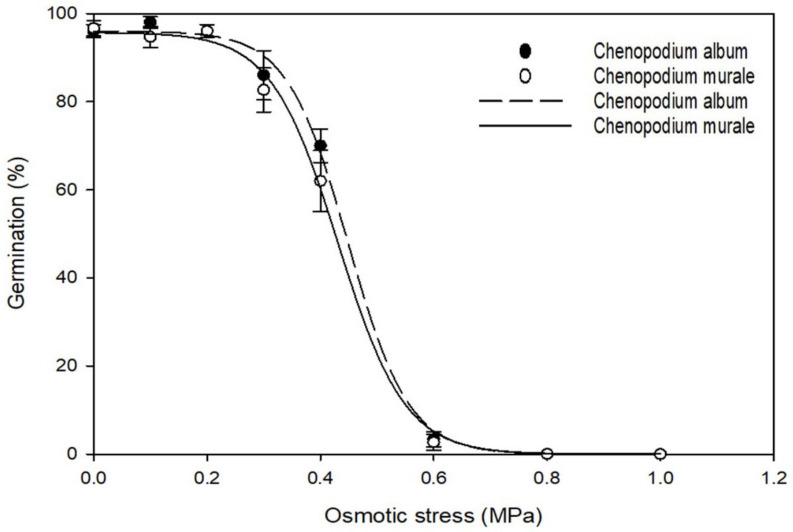
Effect of osmotic stress on germination percentage of *C. album* and *C. murale*.

**Figure 5 biology-11-01599-f005:**
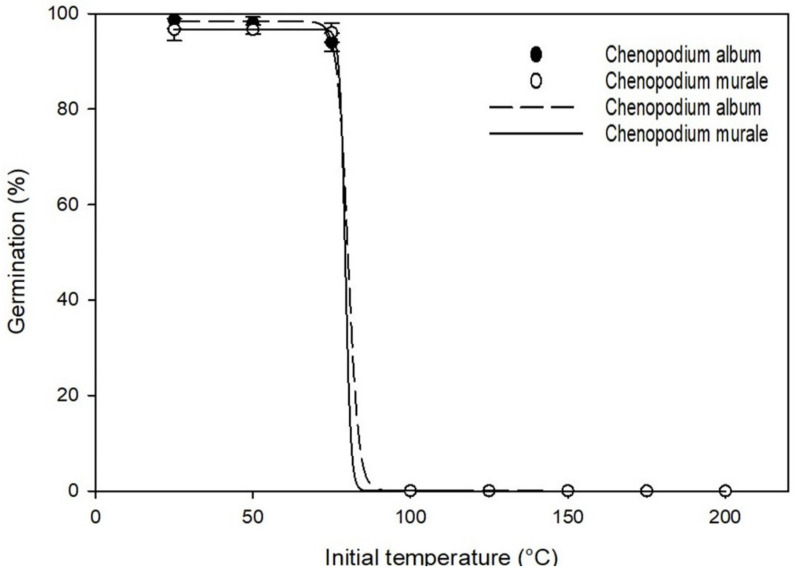
Effect of initial high temperature on germination of *C*. *album* and *C*. *murale*.

**Figure 6 biology-11-01599-f006:**
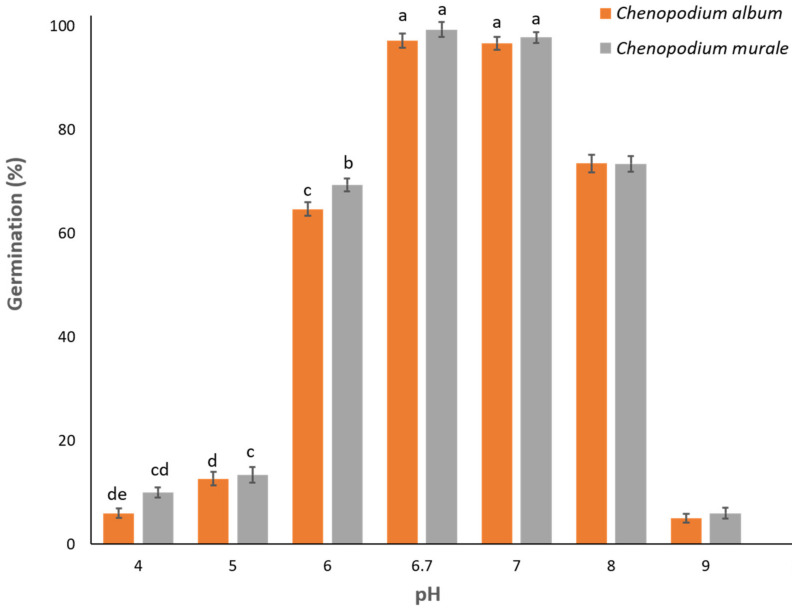
Effect of pH on germination percentage of *C. album* and *C. murale.* Bars with different alphabets are statistically different for a particular species. The species were not compared to each other for germination at different temperatures.

**Table 1 biology-11-01599-t001:** Effect of osmotic stress, salt concentration, and initial temperature on germination behaviour of *C. album* and *C. murale*.

Species	Parameter	Maximum Germination (a)	Germination Rate (b)	GR_50_ * (X_50_)	R^2^	Standard Error of Estimate (S_e_)
*Chenopodium album*	Osmotic stress (MPa)	96	−0.053	0.44	0.99	2.49
	Salt concentration (mM NaCl)	94.1	−22.98	139.89	0.99	2.07
	Initial temperature (°C)	98.3	−1.67	80.15	1.0	0.21
*Chenopodium murale*	Osmotic stress (MPa)	95.7	−0.058	0.43	0.99	2.42
	Salt concentration (mM NaCl)	91.0	−20.32	146.31	0.99	5.25
	Initial temperature (°C)	96.7	−0.87	79.3	1	0.19

* 50% reduction in the maximum germination.

## Data Availability

Not applicable.

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
