# Peer review of "Seed Germination Ecology of Chenopodium album and Chenopodium murale"

_biology, 2022, doi:10.3390/biology11111599_

Round 1
Reviewer 1 Report
The idea of this article is scientifically sound covering “Seed germination ecology of Chenopodium album and Chenopodium murale ” which are important weeds of tropic and subtropic areas. This kind of study is important for the adoption of several management options.
I have concerns about why the authors did not study some direct control strategies like chemical control using recently available herbicides in the market. Also, the light experiment is not well studied and there are missing treatments (24 hr light) that should be included.
Major comments/ suggestions
The abstract needs need more clarity and information.
In the introduction section, there is some scattered information about herbicide resistance without focusing on other weed management options so there is a need for proper structuring by adding some relevant information to support your study.
The research question and objectives should be re-written with mor clarity.
In the material and methods, and results sections, the authors did not mention any information about seed dormancy while your objectives also contain dormancy study.
Also, for light study, there must be a 24-hour light treatment for representing the areas that have a day length of more than 12 hours.
The discussion is also poor and needs to improve with some recent available literature.
For weeds, use full names along with authority at first use and after that use abbreviations and follow them in the whole manuscript. For crops, full names (English and scientific) along with authority at first use and after that use English names throughout the manuscript.
For mentioning units, use proper symbols and follow it whole manuscript.
There is a need for proper formatting while mentioning citations and references.
Minor comments
The genus name must be italic.
L 41: add the word “as well”.
L 46: Need Comma after ‘reaped with crop plants’
L 47: Use “reduce” instead of “reduced”.
L 50-54 Rewrite this sentence and make it clear.
L 100: provide exact information.
L 351, 352, 356 Need proper formatting.
Author Response
Respected Sir/Madam,
Please see the attachment

Reviewer 2 Report
In different sections of the manuscript this parameter is mentioned: GR50. I assume you are referring to "50% germination rate", but please explain the meaning of this acronym and, if appropriate, include a bibliographic citation related to this term.
In section "3.1. Effect of temperature and light on germination" some comments on the effect of light on germination have been included, but there is no table or figure representing the data on the effect of light on germination. Are these data available? Please include them in this section or at least explain where these comments come from.
Table 1 has been included at the end of section 3.1 but refers to section 3.2 and subsequent sections. Please move it to section 3.2 once it has been quoted in the main body of the text.
In table 1, where do the results for "Germination rate (b)" come from?
In the "References" section, the code "doi" has only been included for one reference. Please standardise the style but it is preferable to include the "doi" code for all references that have it.
Author Response

(The authors gave the same response as above.)

Reviewer 3 Report
The work has merit for publication. The manuscript presented a detailed effect of temperature, light, osmotic stress, pH, salinity stress on the weed germination. The results provided more information on the understanding the germination biology of two major weed species under diverse ecological scenarios. However, the research was not deeply. The authors did not develop the mechanism. Further investigations are needed on more variabilities in the future.
Author Response

(The authors gave the same response as above.)

Round 2
Reviewer 1 Report
Authors have sufficiently revised the paper and it is acceptable.